# What Fine-Tuning Changes: A Radiomic Lens on Prostate Foundation Model Representations

**Yipei Wang**[1]                                     YIPEI.WANG@UCL.AC.UK
[1] *UCL Hawkes Institute, Department of Medical Physics and Biomedical Engineering, University College London*

**Yaxi Chen**[1] 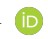                                    YAXI.CHEN.20@UCL.AC.UK
**Wen Yan**[1] 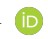                                    WEN-YAN@UCL.AC.UK
**Natasha Thorley**[2,3] 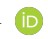                    NATASHA.THORLEY@UCL.AC.UK
[2] *Centre for Medical Imaging, University College London*
[3] *Department of Radiology, University College London Hospital NHS Foundation Trust*

**Alexander Ng**[4] 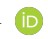                            ALEXANDER.NG@UCL.AC.UK
[4] *Centre for Urology Imaging, Prostate, AI and Surgical Studies (COMPASS) Research Group, Division of Surgery and Interventional Science, University College London*

**Dean C. Barratt**[1] 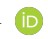                        D.BARRATT@UCL.AC.UK

**Daniel C. Alexander**[1,5] 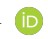              D.ALEXANDER@UCL.AC.UK
[5] *Department of Computer Science, University College London*

**Shonit Punwani**[2,3] 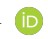                      S.PUNWANI@UCL.AC.UK

**Mark Emberton**[6,7] 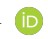                      M.EMBERTON@UCL.AC.UK
[6] *Department of Urology, University College London Hospital*
[7] *Division of Surgery and Interventional Science, University College London*

**Veeru Kasivisvanathan**[4,6] 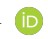          VEERU.KASI@UCL.AC.UK

**Yipeng Hu**[1] 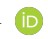                                  YIPENG.HU@UCL.AC.UK

**Editors:** Accepted for publication at MIDL 2026

## Abstract

Clarifying how foundation model encoders change during fine-tuning is important for transparency and trustworthiness in their medical imaging applications. It may also be useful for further understanding, developing, and adapting these models. However, the latent representations produced by such encoders are high dimensional and lack explicit semantic meaning, making it difficult to characterise how task-specific adaptation modifies them. In this study, we introduce a radiomics-based framework that provides an interpretable lens through which these representational changes can be examined and often better understood. Using prostate cancer patient imaging data, we train a two-layer MLP to learn the relationship between radiomic descriptors and encoder embeddings prior to fine-tuning. This model

captures non-linear associations through its first layer, while the final linear layer offers an interpretable mapping from radiomic attributes to (transformed) latent features. To quantify the effect of fine-tuning, the first layer is fixed, and only the linear layer is re-estimated using the embeddings from the fine-tuned encoder. Comparing the pre- and post-fine-tuning linear weights yields a direct quantitative measure of how the encoder's emphasis on specific radiomic characteristics shifts during fine-tuning. We validate the approach using a prostate MRI foundation model and multiple downstream tasks. The analysis reveals consistent, task-dependent changes in the encoder's sensitivity to radiomic texture and intensity features. This work provides the first radiomics-based methodology for systematically interpreting how fine-tuning restructures foundation model representation in medical imaging. The implementation is available at: https://github.com/pipiwang/RadiomicLens.

**Keywords:** mpMRI, Foundation model, Radiomic feature, Interpretability

## 1. Introduction

Foundation models which are pre-trained on large-scale datasets, often using a self-supervised learning paradigm, have increasingly been applied to medical related tasks on various modalities and anatomical structures (Wu et al., 2025; Zhou et al., 2023; Fu et al., 2025; Zhang et al., 2024; McConnell et al., 2025). These recent advances demonstrated the potential of foundation models of achieving better performance for individual applications when being fine-tuned and adapted to specific downstream tasks. However, current research has been largely motivated by and emphasising performance boosts over existing specialised supervised learning models, whereas the interpretability study on foundation models in medical applications is restricted to saliency maps (Zhou et al., 2023) or effects on quantitative results of different fine-tuning strategies (McConnell et al., 2025). The role of fine-tuning in affecting model behaviour and reshaping representation space in foundation models for medical tasks remains insufficiently studied.

Understanding how foundation model encoders change during fine-tuning is essential for interpreting their behaviour in medical image analysis. Although fine-tuning is known to adapt representations towards a target task, the nature of this adaptation is often opaque. Encoder features are high-dimensional and lack clear semantic meaning, which makes it difficult to understand what prior knowledge is retained, what is modified, and how these changes relate to clinically meaningful image characteristics. Despite the recent research efforts on interpreting deep learning models, existing methods face significant limitations. Concept attribution methods such as Testing with Concept Activation Vectors (TCAV) (Kim et al., 2018) require a manually defined set of images that positively represent a concept and another random set that do not; while sparse autoencoders (Cunningham et al., 2023) produce latent dimensions that may correlate with internal representations, but the resulting factors are not guaranteed to be human-meaningful. The discovered concepts often need post-hoc interpretation which would further add the workload of clinical experts in interpreting ambiguous or uninterpretable latent features.

Radiomics provides a complementary and interpretable representation of medical images, consisting of comprehensive descriptions such as texture, intensity statistics, and shape descriptors, for a given region of interest (ROI) (Zwanenburg et al., 2016; Aguirre-Meneses et al., 2025). These features provide quantitative characterisations of medical images which contribute to the radiologists' decision-making process on disease diagnosis and treatment

planning ([Tomaszewski and Gillies, 2021](#)). Unlike the obscure, high-dimensional features from deep learning models, radiomic features have explicit definitions that allow fine-grained interpretation. In this work, we propose to use radiomic features as an interpretable reference space to analyse how a foundation model encoder changes before and after fine-tuning.

This work proposes to model the relationship between radiomic features and encoder embeddings using a two-layer MLP. The network is first trained on the pre-fine-tuning encoder features so that it can learn a general non-linear mapping while keeping the final linear layer directly interpretable. To analyse how fine-tuning alters the encoder representation, we freeze the first layer of the MLP and solve the interpretation linear head using the fine-tuned embeddings. Comparing the two linear heads then highlights how the influence of each radiomic attribute has changed, offering an interpretable view of the effects of fine-tuning.

In this work, we introduce radiomics as an interpretable framework for explaining encoder representations on real patient medical imaging data. We provide the first radiomics-based analysis that makes encoder changes during fine-tuning quantitatively interpretable. The proposed approach is demonstrated using a prostate MRI foundation model across multiple downstream tasks, showing how radiomics reveal task-specific adaptation.

## 2. Method

We introduce a two-stage framework to investigate the relationship between the internal feature representation of the foundation model and radiomic features calculated directly from images, and capture the changes during task-specific fine-tuning, as shown in Fig. 1. At the first stage, features of the foundation model encoder from before and after fine-tuning are projected into a unified, non-linear space; while in the second stage, closed-form linear regression is applied to reconstruct radiomic features from the projected features. By analysing the difference of weights and residuals in the linear decoding between pre- and post- fine-tuning, we provide quantitative measures of representation shift during the fine-tuning process.

### 2.1. Problem formulation

Let $E(\cdot)$ denote a pre-trained foundation model encoder and $E_{ft}(\cdot)$ as one of its fine-tuned variants for a specific downstream task, which both map the input patient data $x$ into a $d$-dimensional feature space. For each patient data $x$, $X^{pre} = E(x)$ and $X^{ft} = E^{ft}(x)$, where $X^{pre} \in \mathbb{R}^d$ and $X^{ft} \in \mathbb{R}^d$ represent the features from the pre-trained and fine-tuned foundation model encoder respectively.

For each patient data $x$, there may exist more than one data modality, for example, various image modality sequences for multi-parametric magnetic resonance imaging (mpMRI). Radiomic features are calculated for each image modality $m$ and the ROI pair, denoted as $R_m \in \mathbb{R}^n, m = 1, 2, ..., M$, where $M$ denotes the total number of modalities of each patient. A set of radiomic features of the same types is calculated for each modality, resulting in $M$ $n$-dimensional vectors, $R = [R_1^\intercal, R_2^\intercal, .., R_M^\intercal]^\intercal \in \mathbb{R}^{nM}$.

The goal is to establish a function

$$f : \mathbb{R}^d \to \mathbb{R}^{nM}, \tag{1}$$

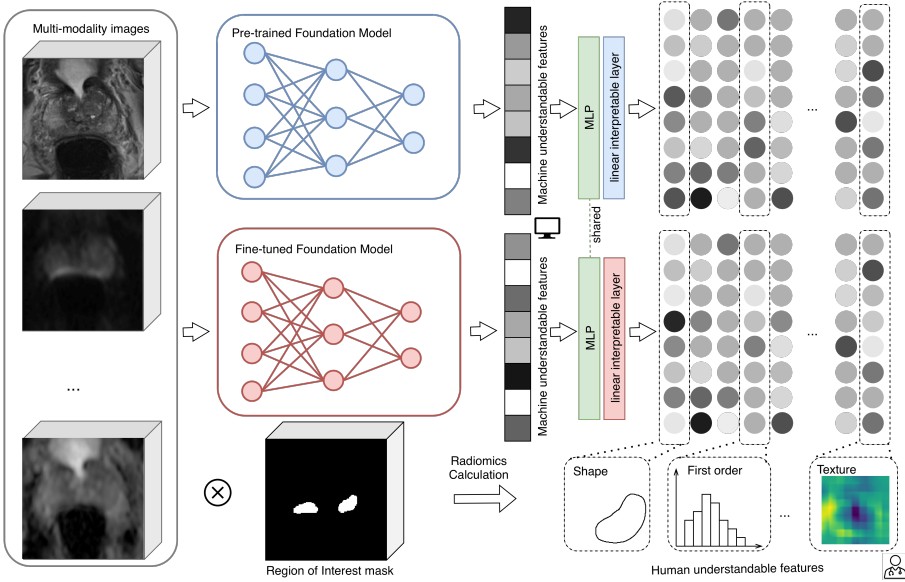

Figure 1: Overview of the proposed framework.

that project the foundation model encoder features to the corresponding radiomic features.

## 2.2. Shared non-linear projection

We first project encoder features into a shared hidden space by learning a shared non-linear function,

$$Z = f^{share}(X; \phi) \tag{2}$$

In order to learn this projection, we decompose $f$ into a shared non-linear projection and modality-specific linear heads, denoted as:

$$f(X) = f^{head}(f^{share}(X; \phi), \theta). \tag{3}$$

The model parameters are optimised using reconstruction loss with optional $L1$, $L2$, or elastic penalties on model weights:

$$\mathcal{L} = \mathcal{L}_{rec} + \lambda_1 ||\phi, \theta||_1 + \lambda ||\phi, \theta||_2^2. \tag{4}$$

Therefore,

$$(\hat{\phi}, \hat{\theta}) = \arg\min_{\phi, \theta} \sum_{m=1}^{M} ||f_m^{head}((f^{share}(X^{pre}, \phi); \theta_m) - R_m||_1 + \lambda_1 ||\phi, \theta||_1 + \lambda ||\phi, \theta||_2^2). \tag{5}$$

The first stage is optimised using features from before fine-tuning as input. Once trained, we freeze the weights of the shared function and discard the temporary heads $f_m^{head}$. The latent features obtained from this stage are denoted as $Z^p = f^{share}(X^{pre}; \hat{\phi})$ and $Z^{ft} = f^{share}(X^{ft}; \hat{\phi})$.

## 2.3. Closed-form linear radiomic decoding

After transforming the features from foundation models to the shared hidden space, we solve a new linear decoder for hidden features from pre- and post- fine-tuning separately using least-squares, in a single matrix form: $R = \theta Z$. The optimal parameters are solved by: $\hat{\theta} = RZ^{\mathsf{T}}(ZZ^{\mathsf{T}})^{-1}$.

## 3. Experiments

### 3.1. Dataset

The proposed radiomic interpretation framework is evaluated on two datasets across three downstream tasks. The first dataset is a multi-study mpMRI collection from the UCL hospital, including SmartTarget (Hamid et al., 2019), PICTURE (Simmons et al., 2018), ProRAFT (Orczyk et al., 2021), Index (Dickinson et al., 2013), PROMIS (Bosaily et al., 2015) and PROGENY (Linch et al., 2017). This dataset contains mpMRI from 850 patients, including T2-weighted image (T2), high-b value diffusion-weighted image (DWI), and Apparent Diffusion Coefficient (ADC) maps, with both lesion and prostate gland masks available on each T2 image. We refer to this dataset as the UCLH dataset in the following sections. The second dataset consists of prostate mpMRI with gland contour from $1,028$ patients recruited in the ReImagine Risk study (Marsden et al., 2021).

All images are resampled to voxel size of $0.5mm \times 0.5mm \times 1mm$. Each dataset is split into train, validation, and test set with a ratio of 7:1:2 on the patient level, for downstream task fine-tuning and evaluation.

### 3.2. Foundation model pretraining

We adopt a recently released prostate mpMRI foundation model, ProFound, to investigate the fine-tuning effect. ProFound uses ConvNeXt v2 (Woo et al., 2023) and is pre-trained using masked autoencoder (MAE) (He et al., 2022), which is a self-supervised learning method that can be used to pre-train vision models. The model is pre-trained on an ensemble of private and public prostate mpMRI datasets of $\sim 5000$ patients, including the PI-CAI dataset (Saha et al., 2024), PROSTATE-MRI from the Cancer Imaging Archive (Choyke et al., 2025), the ReImagine Risk dataset (Marsden et al., 2021) (same as one of the aforementioned downstream dataset), and a private dataset (Min et al., 2025). More details such as pretraining protocols can be found in the ProFound repository [1].

### 3.3. Foundation model fine-tuning

To investigate the feature shift from fine-tuning the pre-trained foundation model, three prostate cancer related downstream tasks were explored.

**Prostate cancer risk group classification.** We fine-tune the ProFound model to predict the Prostate Imaging Reporting and Data System (PIRADS) scores, which is formulated as a multiclass classification task with patient groups of PIRADS scores of $< 3$, 3, 4, and 5. In this task, the ProFound model is fine-tuned by adding a simple head comprising

---

1. https://github.com/pipiwang/ProFound

two fully connected layers with an intermediate batch normalisation layer. The ReImagine risk dataset is used for the classification task and the model is trained with the Cross Entropy loss.

**Prostate cancer lesion segmentation.** The pre-trained model is adapted to delineate the lesion contour by inserting a UPerNet head (Xiao et al., 2018) after the encoder. Dice loss is used for fine-tuning the segmentation task on the UCLH dataset.

**Prostate gland volume estimation.** We also explore a regression task of predicting the prostate gland volume, where a simple head consisting of a batch normalisation layer followed by a fully connected layer is adopted. ProFound is fine-tuned on the UCLH dataset using the Mean Squared Error (MSE) loss.

All fine-tuned models are trained for 100 epochs with a learning rate of 0.001 using the AdamW optimiser. For the classification and segmentation task, the model takes input of three modalities, T2, high-b value DWI, and ADC maps, whereas the regression task only uses T2 as the model input.

### 3.4. Radiomic feature extraction

The radiomic features are extracted following a guidance (Zwanenburg et al., 2016) using PyRadiomics (ver 3.1.0) (Van Griethuysen et al., 2017) on all mpMRI modalities of each patient. A selection of features were extracted, including first-order features, Gray Level Co-occurrence Matrix (GLCM) features, Gray Level Size Zone Matrix (GLSZM) features, Gray Level Run Length Matrix (GLRLM) features, Neighbouring Gray Tone Difference Matrix (NGTDM) features, Gray Level Dependence Matrix (GLDM) features, from both original and Wavelet filtered images. The Min-Max normalisation is performed on each radiomic feature to scale all features to a range of $[0, 1]$.

For the lesion segmentation task, we calculate two sets of radiomics, using lesion masks and prostate gland masks as ROIs respectively. As the data in this study mostly come from patients with low-to-medium-risk cancer, using only the lesion mask would provide a very limited ROI and may exclude regions with subtle extending and/or emerging abnormalities. Including the whole-gland mask would allow us to capture broader contextual and structural information that may be relevant for distinguishing lesion-related changes, especially in early disease. For the PIRADS score classification task and the prostate volume estimation task, we use the whole gland as ROI because PIRADS rely on different primary determining sequence for different zone (peripheral zone and transitional zone), yet using only lesion mask would lose its relative spatial location information.

### 3.5. Training the shared non-linear mapping

To train the shared non-linear mapping $f^{share}$ (in Sect.2.2), we perform a parameter sweep to decide the value of $\lambda_1$ and $\lambda_2$ over the combination of $\lambda_1 \in [0, 10^{-7}, 3 \times 10^{-7}, 10^{-6}, 3 \times 10^{-6}, 10^{-6}]$ and $\lambda_2 \in [0, 10^{-5}, 10^{-4}, 10^{-3}]$. The final adopted values are $\lambda_1 = 10^{-6}$ and $\lambda_2 = 10^{-5}$. The shared non-linear projection mapping is trained for 200 epochs with a learning rate of 0.001.

### 3.6. Representation shift measurement

To quantitatively measure the feature shift in representing radiomics after fine-tuning, we adopt the following metrics.

**Importance score.** For radiomic feature $k$ of modality $m$, the importance is defined as $I_{m,k} = ||\theta_{m,k,:}||_2$. The change after fine-tuning is:

$$\Delta I_{m,k} = I_{m,k}^{ft} - I_{m,k}^{p}. \tag{6}$$

$R^2$ **score.** The $R^2$ score is also known as Coefficient of Determination, which is defined as

$$R^2 = 1 - \frac{Residual\ Sum\ of\ Squares}{Total\ Sum\ of\ Squares}. \tag{7}$$

We use $R^2$ to quantify how well the features from foundation model encoder describes the radiomics information.

**Mean Squared Error**. We also use MSE to investigate the capability of reconstructing radiomic features using the foundation model encoder representation. We performed Wilcoxon signed-rank tests on the per-sample squared errors for each radiomic feature family on the evaluated tasks, comparing the pre-trained model with fine-tuned model.

## 4. Results

### 4.1. Quantitative performance of the interpretation framework

To assess the proposed radiomic feature representation framework, we report MSE and $R^2$ for three downstream tasks and various radiomic feature groups and all the detected statistical significances on squared errors, as shown in Tab. 1. Higher MSE and lower $R^2$ were obtained after the foundation model was adapted to specific tasks, suggesting that feature embeddings become less linearly recoverable into radiomic representations. A possible reason could be that the fine-tuned encoder extracts more specialised information, whereas the self-supervise pre-trained model captures more general information about the input images.

The regression performance of the classification and regression task operated on the prostate gland mask for radiomics calculation. For both downstream tasks, the radiomic representation model exhibited the lowest MSE (0.103 and 0.206) and the highest $R^2$ (0.892 and 0.785) on NGTDM features with and without fine-tuning, indicating information about structural patterns or inflammation relevant markers (Aguirre-Meneses et al., 2025) being encoded by the foundation model. These patterns are likely to contribute more to the model predictive capability for PIRADS score classification and volume regression.

We evaluated the regression performance using radiomic features extracted from two ROI definitions, the lesion mask and the whole prostate gland. When radiomics were computed at the lesion level, the regression achieved low $R^2$, indicating that lesion-level features provide limited explanatory power for the encoder representation in the segmentation task. This is likely due to the high variability and small spatial extent of individual lesions, which produce radiomic descriptors with low stability and weak correlation to global encoder embeddings. In contrast, using gland-level radiomics substantially improved the regression fit, yielding approximately $R^2 \approx 0.4$. This suggests that the encoder captures information more

strongly aligned with global prostate characteristics than with fine-scale lesion appearance. Gland-level attributes such as organ size, zonal anatomy, and age-related tissue changes are known to correlate with cancer presence and localisation, which may explain their stronger correspondence with encoder features. These findings also imply that lesion-specific radiomic features may be too heterogeneous or insufficiently discriminative to establish a stable mapping for this task. For completeness and transparency, we report both ROI scenarios. Together, they illustrate that radiomics extracted at different anatomical scales provide different levels of explanatory value, and that the encoder's behaviour in lesion segmentation is more consistently reflected in global prostate characteristics than in local lesion descriptors.

Table 1: Quantitative results showing the performance of MLP across three downstream tasks on various group of features. CLS = classification task, REG = regression task, SEG-L = segmentation task with lesion mask as radiomics ROI, SEG-P = segmentation task with prostate gland mask as radiomics ROI. Statistical significance levels are denoted by * $p < 0.05$, ** $p < 0.01$, and *** $p < 0.001$.

| Feature | Stage | PIRADS CLS | | Volume REG | | Lesion SEG-L | | Lesion SEG-P | |
|---|---|---|---|---|---|---|---|---|---|
| | | MSE↓ | $R^2 \uparrow$ | MSE↓ | $R^2 \uparrow$ | MSE↓ | $R^2 \uparrow$ | MSE↓ | $R^2 \uparrow$ |
| First order | Pre-train | 0.437 | 0.584 | 0.263 | 0.727 | 0.722 | 0.344 | 0.703 | 0.351 |
| | Fine-tune | 0.452 | 0.569 | 0.366 *** | 0.621 | 0.775 *** | 0.296 | 0.732 ** | 0.325 |
| GLCM | Pre-train | 0.365 | 0.659 | 0.132 | 0.860 | 0.623 | 0.436 | 0.584 | 0.492 |
| | Fine-tune | 0.390 | 0.636 | 0.269 *** | 0.716 | 0.649 ** | 0.412 | 0.613 ** | 0.464 |
| GLDM | Pre-train | 0.326 | 0.697 | 0.162 | 0.826 | 0.625 | 0.440 | 0.593 | 0.468 |
| | Fine-tune | 0.347 * | 0.669 | 0.260 *** | 0.719 | 0.771 *** | 0.307 | 0.620 | 0.442 |
| GLRLM | Pre-train | 0.263 | 0.760 | 0.140 | 0.851 | 0.656 | 0.422 | 0.725 | 0.367 |
| | Fine-tune | 0.331 *** | 0.698 | 0.227 *** | 0.760 | 0.812 *** | 0.272 | 0.746 | 0.348 |
| GLSZM | Pre-train | 0.614 | 0.409 | 0.453 | 0.529 | 0.928 | 0.186 | 0.752 | 0.272 |
| | Fine-tune | 0.631 | 0.393 | 0.577 *** | 0.399 | 1.006 *** | 0.116 | 0.787 *** | 0.238 |
| NGTDM | Pre-train | 0.287 | 0.735 | 0.103 | 0.892 | 0.712 | 0.361 | 0.665 | 0.400 |
| | Fine-tune | 0.300 ** | 0.717 | 0.206 *** | 0.785 | 0.838 *** | 0.263 | 0.735 *** | 0.336 |
| Original | Pre-train | 0.416 | 0.602 | 0.640 | 0.335 | 0.691 | 0.311 | 0.641 | 0.368 |
| | Fine-tune | 0.453 * | 0.571 | 0.747 *** | 0.224 | 0.810 *** | 0.192 | 0.687 *** | 0.322 |
| Wavelet | Pre-train | 0.430 | 0.599 | 0.279 | 0.704 | 0.791 | 0.302 | 0.776 | 0.308 |
| | Fine-tune | 0.441 | 0.585 | 0.354 *** | 0.623 | 0.837 ** | 0.259 | 0.792 | 0.293 |

In addition to reporting radiomics group-wise performance, we also investigate the regression fit on different input modalities. We show three examples in Fig. 2, including the PIRADS classification task and the segmentation task with two ROI definitions. Only the feature types with the highest $R^2$ after fine-tuning are illustrated as examples. It can be observed that in the PIRADS classification task, radiomics of all three modalities are aligned to the foundation model features to a substantial degree, indicating comparative predictive power of all modalities. While for the lesion segmentation task, high-b value DWI and ADC radiomics align with foundation model features while T2 bahaves the opposite, regardless of pre- or post- fine-tuning, which corresponds to the ability of ADC and high b DWI in

identifying tumours by highlighting cancerous tissues with signal intensity different from normal tissues.

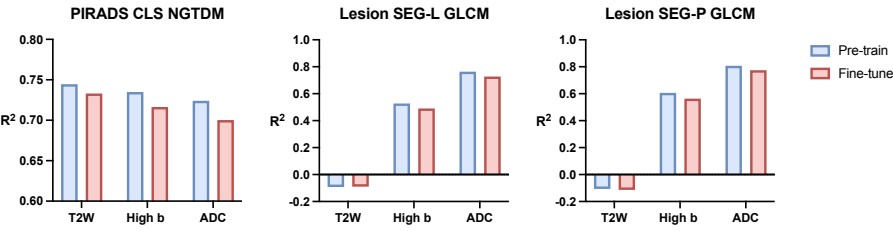

Figure 2: $R^2$ of single modality on mpMRI tasks on the highest overall $R^2$ features.

## 4.2. Qualitative case studies

To understand how fine-tuning reshapes the encoder representation for PIRADS prediction, we examined radiomic features whose ability to explain encoder embeddings changed the most, as shown in Fig. 3. For each radiomic descriptor, we quantified its explanatory value via $\Delta I$ calculated before and after fine-tuning, then ranked the features in three complementary ways: those that became newly important (positive $\Delta I$), those whose importance decreased (negative $\Delta I$), and those with the least absolute change among features that were globally stable. These three views reveal how fine-tuning shifts the encoder toward radiologically meaningful image characteristics used in PIRADS scoring.

**Features gaining importance: increased alignment with T2 structural patterns.** Radiomic features with the largest positive $\Delta I$ were predominantly T2-weighted texture and run-length measures, including GLCM-MCC, GLCM-correlation, and several GLRLM descriptors capturing long high-intensity runs or low-intensity structural streaks. These features quantify the coherence, organisation and zonal architecture of the prostate, for example in the peripheral and transition zones. PIRADS guidelines place a strong emphasis on these T2 features, as lesion visibility and morphological distortion on T2-weighted imaging are central markers of clinical suspicion. The increased ability of these descriptors to explain the encoder after fine-tuning indicates that the model becomes more attuned to the structured appearance of malignant regions and to deviations from normal glandular texture patterns. This shift suggests that fine-tuning makes the encoder more aligned with radiologists' use of T2 structural cues for PIRADS assessment.

**Features losing importance: reduced reliance on unstable diffusion heterogeneity.** Radiomic features with the strongest negative $\Delta I$ primarily originated from ADC and high-b diffusion imaging. These include ADC-GLCM-Imc2, ADC-GLCM-DifferenceEntropy, ADC-GLDM-DNUN, and high-b kurtosis, all of which characterise fine-scale texture irregularity and local signal heterogeneity. While such patterns may loosely correlate with tumour presence, they are also known to be sensitive to acquisition noise, variation in b-values and small ROI instability, and they are not central to PIRADS scoring, which emphasises consistent diffusion restriction rather than stochastic heterogeneity. The drop in explanatory power after fine-tuning suggests that the encoder moves away from

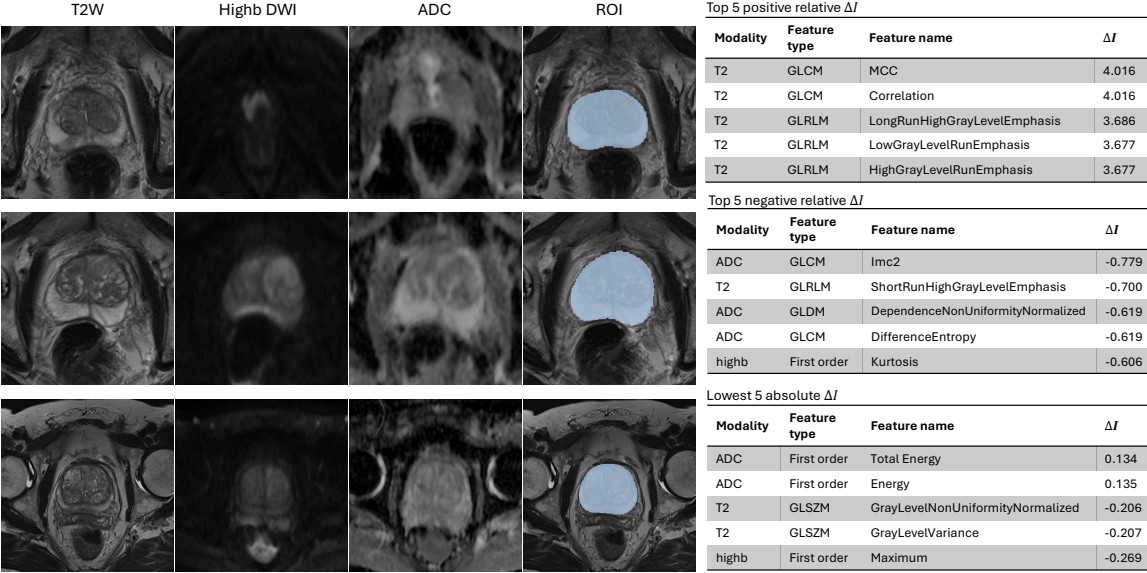

Figure 3: Radiomic feature changes for PIRADS classification tasks. The left side shows patient cases with all modalities and the prostate gland ROI, and right side lists features with the top and least five $\Delta I$.

these volatile cues, instead prioritising more robust diffusion features. This behaviour reflects the goal of fine-tuning: to refine the model toward radiologically reliable indicators of high PIRADS scores and to suppress spurious correlations.

**Features with low shifts: stable global ADC and T2 indicators.** Among radiomic features that explained a substantial portion of encoder variance both before and after fine-tuning, those with the lowest absolute changes while maintaining moderate to high importance score included global ADC intensity descriptors (TotalEnergy, Energy) and T2 gray-level distribution measures (GLSZM-GLNUN, GLSZM-Variance). These metrics quantify overall diffusion signal level and non-uniformity, as well as global heterogeneity across the gland. These are radiologically meaningful biomarkers: PIRADS scoring heavily relies on diffusion restriction (low ADC) and lesion conspicuity cross-correlated across T2 and diffusion sequences. These stable features correspond to fundamental anatomical or textural characteristics that provide contextual structure but are not strongly task-discriminative for PIRADS scoring.

Using our proposed approach, across all the analyses, fine-tuning produces a systematic, clinically interpretable reorganisation of encoder features. The encoder strengthens its alignment with zonal T2 textural structure, reduces sensitivity to noise-prone diffusion heterogeneity, and retains the weighting of global ADC/T2 intensity and non-uniformity—all of which mirror established criteria in PIRADS scoring. These results show that radiomics provides a meaningful, interpretable lens through which fine-tuning behaviour can be un-

derstood, demonstrating that the model becomes progressively more "radiologist-like" in the cues it uses to assess prostate cancer suspicion.

## 5. Discussion

Radiomic features are known to be correlated or measure overlapping aspects of image texture and intensity, however, in this specific study, our goal is not to construct a minimal or decorrelated feature set, but to use these features as descriptive markers to monitor distributional shifts before and after the fine-tuning process. In this context, correlation among features does not diminish the value of the intended objectives, because it is not expected to use these features for predictive modeling or inference. Instead, any feature that reflects a change induced by fine-tuning is informative for our purpose. To mitigate potential instability arising from collinearity, we incorporate both $L1$ and $L2$ regularization within our methods.

Radiomic features also vary in their degree of interpretability. While some descriptors are less intuitive at a clinical level, they are still substantially more interpretable than most deep learning features, which are typically abstract latent representations without any human-interpretable incentives. Future work could benefit from identifying and selecting a more interpretable subset of radiomic features for characterising fine-tuning effects.

We acknowledge that the presence of radiomic features in the embeddings does not guarantee their usefulness for downstream tasks. Therefore, we also report downstream results for both 1) models with a frozen backbone and trained task-specific head, and 2) models with both backbone and head fine-tuned in the Appendix. These correspond directly to the pre- and post- fine-tuning embeddings used in our radiomic analysis, and enable a more direct link between representation changes and downstream task performance. Notably, across all evaluated tasks, fine-tuning consistently yields improved downstream performance.

All experiments in this study were conducted using ProFound, which served as a concrete example for applying our radiomic analysis framework. While our methodological approach is model-agnostic and, in principle, applicable to any foundation model, we acknowledge that the specific feature-level findings reported here are likely influenced by the particular pretraining data, architecture, and downstream tasks used. As such, we do not claim that specific feature-level conclusions are expected to generalise further. For example, fine-tuning effects may vary for models pretrained on prostate MR images versus chest CT images, or for tasks such as segmentation versus classification. Nevertheless, although a much larger-scale study is required, understanding these differences or identifying potential feature shifts that remain invariant to pretraining, domain, or task remains an important future direction.

## 6. Conclusion

This study introduces a radiomics-based framework for interpreting how foundation model representations change during fine-tuning, offering a quantitative and clinically meaningful lens on encoder behaviour. Across three downstream prostate MRI tasks, we showed that radiomic features can partially reconstruct encoder embeddings, with the best alignment observed for, in particular, at gland-ROI-level radiomics and for modality–feature combi-

nations consistent with radiological practice (e.g., high-b and ADC for lesion identification, global features for cancer risk). By comparing linear decoding weights before and after fine-tuning, our method revealed systematic and task-dependent shifts, for example increased sensitivity to zonal T2 structure for PIRADS prediction, reduced reliance on unstable diffusion heterogeneity and stable weighting of global ADC/T2 signal characteristics.

While the radiomics-to-latent mapping explains only a fraction of encoder variance, this limitation reflects the complementary nature of hand-crafted radiomics and high-capacity foundation models. Future work may explore richer interpretable feature spaces, non-linear or sparsity-aware decoding and cross-task comparisons to establish a broader taxonomy of fine-tuning-induced representation changes. Overall, our findings demonstrate that radiomics provides a practical and principled interpretability tool, enabling foundation models to be probed, compared, adapted and potentially better understood with greater transparency in medical imaging applications.

## Acknowledgments

This work was supported by the National Institute for Health Research (NIHR) University College London Hospitals (UCLH) Biomedical Research Centre (BRC). This work was also supported by the International Alliance for Cancer Early Detection, an alliance between Cancer Research UK [C28070/A30912; C73666/A31378], Canary Center at Stanford University, the University of Cambridge, OHSU Knight Cancer Institute, University College London and the University of Manchester.

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

# Appendix A. Appendix

Table 2: Downstream task results for both models where the backbone is frozen and only the task-specific head is trained (named after "Pre-trained" below), and models where both the backbone and head are fine-tuned (named after "Fine-tuned"). CLS = classification task, REG = regression task, SEG = segmentation task, QWK = Quadratic Weighted Kappa.

|  | PIRADS CLS | Volume REG | Lesion SEG |
| --- | --- | --- | --- |
|  | QWK ↑ | MSE ↓ | Mean Dice ↑ |
| Pre-trained | 0.118 | 0.007 | 0.399 |
| Fine-tuned | 0.326 | 0.004 | 0.429 |

