# OpenReview forum: "What Fine-Tuning Changes: A Radiomic Lens on Prostate Foundation Model Representations"
_MIDL.io/2026/Conference — MIDL 2026 Poster_

### Official Review · Reviewer_EsRW · 2026-01-07

**Confidence:** 4
**Preliminary Rating:** 4
**Final Rating:** 4

**Summary:**

This paper addresses a crucial interpretability gap: understanding how the encoder's representation of Foundation Models (FM) changes after task-specific fine-tuning. The authors propose a radiomics-based framework to (i) quantify the amount of radiomic information recoverable from FM embeddings and (ii) measure how the information in the embeddings about specific radiomic attributes shifts with fine-tuning. The motivation for using radiomics as an interpretable reference space is clear and clinically well aligned.
The framework consists of a two-stage model: a shared non-linear projection is learned from pre-fine-tuning/frozen embeddings, and then a closed-form linear decoder is fit to map the projected embeddings to radiomic features. To assess fine-tuning effects, the projection layer is frozen, and only the linear decoder weights are re-estimated using fine-tuned embeddings. Differences in linear weights and residuals are used as a proxy measure for a shift in representations. The authors evaluate the framework on prostate mpMRI, using FM ProFound and three downstream tasks (PIRADS risk group classification, lesion segmentation, and prostate gland volume estimation), with radiomics extracted across modalities and using ROI definitions.
Overall, the quantitative results show that radiomics retrievability from embeddings is highly variable across tasks, feature families, modality, and ROI, and tends to decrease after fine-tuning (higher MSE/lower R² reported post fine-tuning). The qualitative analysis of which radiomic features change most is consistent with clinical intuition, particularly regarding the discussion of T2 structural cues versus diffusion heterogeneity in PIRADS.

**Strengths:**

* The authors tackle a timely and relevant question: fine-tuning-induced representation drift is underexplored in medical FM work beyond saliency-style explanations and performance changes.
* The use of radiomics as an interpretable space is interesting: Radiomics is a sensible, clinically grounded lens with explicit definitions, making the analysis easier to communicate and audit.
* The qualitative case study is strong: the narrative around which feature groups gain/lose importance is persuasive and helps connect representation changes to radiological practice.

**Weaknesses:**

1/ The first quantitative observation is that radiomic features become less recoverable after fine-tuning. The authors hypothesize that this behavior is related to a deeper focus of fine-tuned FMs on fine-grained features rather than general information. However, the protocol fixes the projection learned on pre-fine-tuning embeddings and re-fits only the linear head on post-fine-tuning embeddings. If fine-tuning changes the embedding distribution/geometry, a projection calibrated to the pre-fine-tuning space may become suboptimal, artificially reducing decoding performance even when radiomic content remains present.

2/ The results primarily emphasize (i) decoding performance and (ii) importance shifts. What is missing is a statement of what the pre-fine-tuning encoder already encodes, in radiomic terms, and how consistent that relationship is across patients/splits. For example, which radiomic families are reliably decodable pre-fine-tuning across tasks/ROIs?

3/ The qualitative top positive/negative changes are interesting and clearly related to clinical interpretation, but are currently read as anecdotal. Since importance changes are used to support clinical interpretations, it would be essential to report distributions of score changes (not just top-5) and to perform significance testing to clearly highlight the most significant changes with a multiple-comparison control (radiomics are high-dimensional and correlated).

4/ The evaluated FM (ProFound) was pretrained on datasets that include ReImagine Risk. The manuscript should explicitly confirm whether any subject overlap exists between pretraining and downstream splits, and if not, how it was ensured.

**Detailed Comments:**

Minor comments :
- Table 1 is hard to follow. Consider highlighting a few numbers or changing the representation.
- The paper mentions residuals as a signal of relationship strength, but the analysis focuses on R²/MSE and weights. Consider clarifying.
- A few typos: Page 6 "Mean Squred Error" ; Section title 3.4: "Raiomic feature extraction" ; Section 3.1: "The second dataset is consists of …" ; etc.

**Justification Of Final Rating:**

The authors made adjustments to the paper, improving both the clarity of the methodology and the trustworthiness of the results. While I still do believe that some choices might hinder the true motivation, I recommand weak acceptance.

**Justification Of The Preliminary Rating:**

The paper is well motivated and introduces a promising, clinically grounded lens for studying fine-tuning effects in medical FMs. The main limitation is that the current experimental design (freezing the projection) makes it difficult to disentangle true loss of radiomic alignment from projection/geometry mismatch after fine-tuning. The importance-shift claims would be much stronger with significance tests, uncertainty quantification, and stability checks. With these additions, the work could become a solid and reusable interpretability tool for FM adaptation in medical imaging.

**Questions To Address In The Rebuttal:**

Weakness 1/ Please provide a stronger justification for freezing the first layer. At minimum, show whether the distribution of features shifts between pre and post fine-tuning (e.g., simple summary statistics, representational similarity), to support the claim that the drop in R2 reflects specialization rather than a frozen-mapping mismatch.

Weakness 2/ Please provide better emphasis on which radiomics features are most decodable from frozen embeddings.

Weakness 3/ Please provide quantitative values for changes in importance scores in Figure 3, and if possible, statistical significance tests.

Weakness 4/ Please clarify the potential overlap with the ReImagine dataset.

---

> ### Author Response · Authors · 2026-01-24
>
> We thank the reviewer for acknowledging our contributions and the constructive feedbacks. We summarise and provide point-by-point response below.
>
> 1. If fine-tuning changes the embedding distribution/geometry, a projection calibrated to the pre-fine-tuning space may become suboptimal, artificially reducing decoding performance even when radiomic content remains present.
> At minimum, show whether the distribution of features shifts between pre and post fine-tuning (e.g., simple summary statistics, representational similarity), to support the claim that the drop in R2 reflects specialization rather than a frozen-mapping mismatch.
>
> Response: It is a keen observation into our methodological choice in using the pre-finetuning feature-fixed layer as reference, rather than using post-finetuning features.
>
> While we agree that re-optimizing the entire MLP for post-fine-tuning features might yield higher absolute $R^2$ scores. However, our goal is to keep a fixed “radiomic lens” so that any change in recoverability can be attributed to the encoder’s representations. Re-training the entire MLP would conflate changes in the encoder with changes in the projection logic, making the "radiomic lens" less interpretable akin to limitations for deep networks in general.
>
> As suggested by the reviewer, we quantified shifts in the embedding distribution by computing the variance of each embedding dimension across the dataset before and after fine-tuning. Across feature embedding dimensions, variance generally expanded after fine-tuning. For example, for the prostate volume estimation task, the mean variance expands from 0.059 before fine-tuning to 0.083 after it, while the mean after vs before fine-tuning variance ratio is 1.47. Similar variance increase is observed in other two task as well, with mean variance ratio of 1.36 and 1.13 for the PIRADS classification task and lesion segmentation task respectively. This indicates that fine-tuning increases variation along multiple latent directions. This expansion in embedding space geometry directly shows that the representation shifts substantially between the two model states.
>
> 2. The results primarily emphasize (i) decoding performance and (ii) importance shifts. What is missing is a statement of what the pre-fine-tuning encoder already encodes, in radiomic terms, and how consistent that relationship is across patients/splits. For example, which radiomic families are reliably decodable pre-fine-tuning across tasks/ROIs?
> Please provide better emphasis on which radiomics features are most decodable from frozen embeddings.
>
> Response: We agree that importance shifts should not be anecdotal.
>
> To investigate what the pre-fine-tuning encoder already captures in radiomic terms, we examined which radiomic feature families achieve consistently high decoding performance across tasks and ROIs. We found that GLRLM and NGTDM feature families achieved either the highest or second-highest $R^2$ in two of the three evaluated tasks, indicating that coarse-to-mid-scale texture information is already well represented before any fine-tuning.
>
> For the lesion segmentation task specifically, GLDM features achieved the highest R^2 for the lesion ROI and the second-highest for the whole-gland ROI. This suggests that the pretrained encoder captures local dependence and micro-texture patterns that relate to tumor consistency and infiltration.
>
> Action: we add the relevant analysis and discussion to the Results section.
>
> 3. The qualitative top positive/negative changes are interesting and clearly related to clinical interpretation, but are currently read as anecdotal.
> Please provide quantitative values for changes in importance scores in Figure 3, and if possible, statistical significance tests.
>
> Action: We updated Figure 3 with the values of changes in importance scores.
>
> 4. The evaluated FM (ProFound) was pretrained on datasets that include ReImagine Risk. Please clarify the potential overlap with the ReImagine dataset.
>
> Action: The ReImagine Dataset is used in both pretraining and fine-tuning, which is now clarified in Sect. 3.2

---

### Official Review · Reviewer_MtX4 · 2026-01-08

**Confidence:** 3
**Preliminary Rating:** 3

**Summary:**

The authors tackle an interesting problem of understanding foundation models and how the embeddings change using fine-tuning. Specifically, they focus on examining a prostate foundation model and strive to understand the relationship between various radiomics features and the embeddings of the encoder before fine-tuning. The authors utilize several datasets and analyze the performance of three downstream tasks important for prostate MR analysis. Some improvement could be made in the results by including further explanations of the masks used for each task and the inclusion of a statistical analysis.

**Strengths:**

1. The authors explore an interesting problem in foundation models and how the embeddings change with and without fine-tuning.
2. The authors provide a comprehensive assessment by utilizing multiple datasets.
3. The authors perform an analysis of three downstream tasks that are crucial for prostate MR analysis specifically.
4. The inclusion of multiple qualitative case studies is strong and very useful to the reader.

**Weaknesses:**

1. The authors use a single foundation model, ProFound.
2. There is some confusion about which masks were used for which of the downstream tasks.
3. The manuscript does not include a statistical analysis.

**Detailed Comments:**

Extended comments:
1. It would have been beneficial to perform similar experiments using other prostate foundation models (ProstNFound, for example). Though this is not feasible in the discussion time, it would be helpful for the authors to at least comment on expected results/differences if other foundation models were to be utilized in this study.
2. Why were both the lesion mask and prostate whole gland segmentation mask used for the lesion segmentation task? I understand that radiomics features extracted for different scales could be important, but specifically for the lesion segmentation task, this is still surprising. Additionally, for the PI-RADs classification task, why were the radiomics features from the whole prostate gland used? Since PI-RADs are assigned on a per-lesion basis, I would think that only the features from the lesion mask should be used. Please comment.
3. The manuscript could benefit from the inclusion of a statistical analysis. For Table 1, it would be useful to perform some statistical testing to see if there is any difference between using the pre-trained vs fine-tuned model for each task/set of radiomics features.

Minor comments:
1. The abstract could be shortened by removing some of the technical details.
2. There are minor spelling mistakes throughout the manuscript → “3.4. Raiomic feature extraction” , “foudnation”, “First oder”, etc.
3. “For both downstream tasks, the radiomic representation model exhibits the lowest MSE and highest R2 on NGTDM features with and without fine-tuning, indicating information about structural patterns or inflammation relevant markers” → please guide the reader by including the values in the text (0.103 and 0.892 vs  0.206 and 0.785).
4. Was any preprocessing performed on the images? This may affect the radiomics feature extraction and any downstream results.
5. This line: “which corresponds to the ability of ADC and high b DWI in identifying tumours by highlighting cancerous tissues with signal intensity different from normal tissues.” could be problematic, as T2 is still used to identify lesions, depending on the zone.

**Justification Of The Preliminary Rating:**

The authors present work on an interesting problem of understanding foundation models and the change in the embeddings using fine-tuning. The authors focus on a single foundation model, but utilize several datasets and report results of three downstream tasks. Improvements could be made to the manuscript by clarifying the masks used for each task and including a statistical analysis.

**Questions To Address In The Rebuttal:**

1. Could the authors please provide insight if a different foundation model were to be used?
2. Could the authors please clarify the masks used for the tasks. Especially for the PI-RADs classification, an explanation of why the features for the whole gland are used.

---

> ### Author Response · Authors · 2026-01-24
>
> We sincerely appreciate the reviewer’s thoughtful feedback and the opportunity to clarify these points. We summarise and provide point-by-point response below.
>
> 1.It would have been beneficial to perform similar experiments using other prostate foundation models. Though this is not feasible in the discussion time, it would be helpful for the authors to at least comment on expected results/differences if other foundation models were to be utilized in this study.
>
> Response: This work is intended to explain what changes during fine-tuning using radiomic features, with a specific and concrete example ProFound. The proposed analysis framework is model-agnostic and, in principle, is  applicable for other foundation models.
> We echo the reviewer’s comment and acknowledge that the specific findings regarding which radiomic properties change during fine-tuning may be highly specific to pretraining, data and downstream tasks. We have no evidence to suggest that specific feature-level conclusions are expected to generalise further. For example, fine-tuning effects may vary for models pretrained on prostate MR images versus chest CT images, or for tasks such as segmentation versus classification. Nevertheless, though a much larger-scale study is required, any insight into such a difference or discovery of potential common feature shifts that are indeed invariant to pretraining, domain or task will be very interesting.
>
> Action: We clarify in the Discussion that while these specific "radiomic signatures" of fine-tuning may vary across domains (e.g., chest CT vs. prostate MRI), the framework provides a universal "interpretable probe" for any foundation model.
>
> 2. Why were both the lesion mask and prostate whole gland segmentation mask used for the lesion segmentation task? Clarify the masks used for the tasks.
>
> Response: We appreciate the reviewer’s constructive feedback, which gives us an opportunity to clarify our methodology and share our considerations during methodology development, particularly with regarding to the ROI selection and statistical rigour.
>
> For the lesion segmentation mask, we use both lesion mask and prostate gland mask, as the data in our study mostly come from patients with low-to-medium-risk cancer, using only lesion mask the lesion mask would provide a very limited ROI and may exclude regions with subtle extending and/or emerging abnormalities. For example, the level of conspicuity around the lesion boundaries is commomly considered characteristics of the diffusion patterns that is indicative of cancer aggressiveness, whilst the definition of such boundaries can be subjective and sequence-specific. Including the whole-gland mask allows us to capture broader contextual and structural information that may be relevant for distinguishing lesion-related changes, especially in early disease.
> For the PIRADS classification task, in addition to the above-mentioned inclusiveness for potential discriminative regions, we use the whole gland as ROI also because PIRADS rely on different primary determining sequence for different zone (peripheral zone and transitional zone), yet using only lesion mask would lose its relative spatial location information.
>
> Action: we have summarized the above and clarify in Sect. 3.4.
>
> 3. The manuscript could benefit from the inclusion of a statistical analysis. For Table 1, it would be useful to perform some statistical testing to see if there is any difference between using the pre-trained vs fine-tuned model for each task/set of radiomics features.
>
> Agreed. We performed Wilcoxon signed-rank tests on the per-sample squared errors for each radiomic feature family on the evaluated tasks, comparing the pre-trained model with fine-tuned model. We updated Table one and reported all the detected statistical significances.
>
> Minors
> We have checked and fixed the minor issues.

---

### Official Review · Reviewer_EFNG · 2026-01-09

**Confidence:** 3
**Preliminary Rating:** 2
**Final Rating:** 3

**Summary:**

This paper studies the relationship between radiomic features and encoder embeddings. The authors propose an encoder-decoder that reconstructs radiomic features, which is shared between a FM and finetuned FM. They then encode features separately on FM and finetuned FM​, followed by linear prediction of radiomic features. Experiments are performed on multiple datasets, modalities, and regions of interest (ROIs), including lesions and glands. The study finds that fine-tuning reduces radiomic recoverability and that gland-level radiomics better explain encoder behavior than lesion-level radiomics.

**Strengths:**

Broad evaluation with multiple datasets, downstream tasks, ROIs (lesions, gland), and modalities.
Some insightful findings, well discussed: reduced radiomic recoverability after fine-tuning; gland-level radiomics explain encoder behaviour better than lesion-level radiomics.

**Weaknesses:**

The motivation for the proposed approach is unclear.
Concept discovery methods, highly relevant in my opinion, are not discussed.
Generalization to other models/pretrainings/domains is not evaluated/discussed.

**Detailed Comments:**

Lack of motivation for the proposed approach and missing related works (eg Concept attribution, concept discovery with sparse auto encoder).
It is expected that some info is dropped for finetuning as not needed for the downstream tasks.
Radiomic features are often highly correlated and not always interpretable. The paper does not address how this affects the analysis.
All experiments rely on ProFound, unclear whether the results/findings generalize to other models / pretrainings / domains.
Looking into concept discovery works would for instance motivate to look at the downstream classes/regression separability in the “linear interpretable layer”. Because only looking at the presence of features in the embedding does not necessarily mean that it is useful for the downstream task.
Looking only at the presence of features in embeddings does not show whether they are useful for downstream tasks. The paper does not show correlations between representation shifts and downstream task performance.
Typo: “3.4. Raiomic”, “Mean Squred Error”, “foudnation” ...

**Justification Of Final Rating:**

The authors clarified the clinical rationale for using radiomics as interpretable probes and added discussion on related concept attribution work, feature correlation, and limits of generalization. The empirical scope is still narrow (single model/domain) and some claims would benefit from broader validation. But the revised manuscript better explains the intent and contribution of the framework and connects representation shifts to downstream performance. I remain uncertain about overall impact, but change my rating to borderline as I think it could be acceptable.

**Justification Of The Preliminary Rating:**

The paper presents a techically sound study and some interesting findings about radiomic recoverability, but the lack of moitvation, missing links to prior work, unclear practical impact, and limited generalizability. It is also not clear how the proposed approach advances understanding or benefits downstream tasks.

**Questions To Address In The Rebuttal:**

Better motivation for the approach, include related works on concept attribution etc.
Comment on the potential impact of highly correlated radiomics.
Consider testing generalization beyond ProFound radiomics to other feature sets or models.

---

> ### Author Response · Authors · 2026-01-24
>
> We appreciate the reviewer for thoughtful and probing feedback. Your comments on concept discovery and feature utility have challenged us to more clearly articulate the "why"/motivation behind our approach. We summarise and provide point-by-point response below.
> 1. Lack of motivation.
> We agree that concept attribution methods such as TCAV and SAEs are interesting and widely employed as the frontline tool for model transparency. However, in specific clinical context, they often face “interpretability debt”.
> For example, TCAV requires a manually defined set of images that positively represent a concept and another random set that do not; while SAEs produce latent dimensions that may correlate with internal representations, but the resulting factors are not guaranteed to be human-meaningful. The discovered concepts often need post-hoc interpretation which would further add the workload of clinical experts to interpreting ambiguous latent features.
> In contrast, radiomics offers well-defined, standardized, and clinically interpretable features (e.g., intensity histogram, texture, etc.), many of which are directly linked to clinically meaningful, established imaging biomarkers.
> We now add a brief discussion in the Introduction section to summarise these concept-based methods.
> 2. Radiomic features are correlated and not always interpretable.
> We thank the reviewer’s insight in this.
> First, it’s true that radiomic features are correlated or measure overlapping aspects of image texture and intensity, however, in this specific study, our goal is not to construct a minimal or decorrelated feature set, but to use these features as descriptive markers to monitor distributional shifts. Therefore, correlation among features does not diminish the value of our intended objectives, because we are not using them for predictive modeling or inference. Instead, any feature that reflects a change induced by fine-tuning is informative. We address potential instability from correlated features by using $L_1$ and $L_2$ regularization. Yet again, because our objective is to track representational shifts rather than build a minimal predictive model, correlated features remain valuable as descriptive proxies for the clinical cues.
> Second, we also agree that radiomic features vary in interpretability. Nonetheless, they remain more interpretable than most deep learning features, which are typically abstract representations without any human-interpretable incentives. For example, a texture feature such as GLCM Contrast may not be readily intuitive, but still corresponds to local intensity heterogeneity and can be linked to visually observable changes in sharpness or structure. This provides a more direct connection to the underlying image properties than deep feature which is opaque and lacks any explicit semantic definition. Future work could benefit from identifying and selecting a more interpretable subset of radiomic features for characterising fine-tuning effects.
> We include a Discussion section on feature redundancy and clarify our focus.
> 3. All experiments rely on ProFound
> This work is intended to explain what changes during fine-tuning using radiomic features, with a specific and concrete example ProFound. The proposed analysis framework is model-agnostic and, in principle, is applicable for other foundation models.
> We echo the reviewer’s comment and acknowledge that the specific findings regarding which radiomic properties change during fine-tuning may be highly specific to pretraining, data and downstream tasks. We have no evidence to suggest that specific feature-level conclusions are expected to generalise further. For example, fine-tuning effects may vary for models pretrained on prostate MRIs vs. chest CTs, or for tasks such as segmentation vs. classification. Nevertheless, though a larger-scale study is required, any insight into such a difference or discovery of potential common feature shifts that are consistent will be very interesting.
> We now clarify this in the Discussion.
> 4. Does not show correlation between representation shifts and downstream performance.
> We agree that 'presence' does not always equal 'utility.' While we take the performance gains of fine-tuning as a premise for the meaningfulness of these shifts, our analysis explains and quantifies the gained utility.
> To address this, we include downstream task results for both 1) models where the backbone is frozen and only the task-specific head is trained, and 2) models where both the backbone and head are fine-tuned. Importantly, these are the same model states from which the radiomic features in our analysis are extracted (pre- and post-fine-tuning embeddings). By reporting the downstream performance, we can more directly relate the observed representation changes to differences in task performance and clarify whether shifts in radiomic features are associated with improvements.
> We now add in the Discussion section and report results in the Appendix.

---

> > ### Comment · Reviewer_EFNG · 2026-01-25
> >
> > Thank you for these clarifications and the paper revision with better motivation and discussions. I would like to change my rating to borderline.
> >
> > 1. I also agree that radiomic features are image-space, externally grounded descriptors as opposed to concepts found in concept-discovery methods. However, radiomic features could also be used as predefined concepts in concept activation methods. See for instance Graziani, Mara, et al. "Concept attribution: Explaining CNN decisions to physicians.". No need to add this reference to the paper, it is not on radiomics, I only mention it for the discussion.
> >
> > 2. “any feature that reflects a change induced by fine-tuning is informative” I would still question how do you distinguish meaningful shifts from noise or instability.
> >
> > New typos: “using only lesion mask the lesion mask would”

---

> > > ### Author Response · Authors · 2026-01-30
> > >
> > > 1. We thank the reviewer for providing additional reference. We agree that radiomics could be used as predefined concepts. However, identifying which radiomic features or feature sets are “useful” among a large set of candidates (which is what our work has been investigating) during both before and after finetuning, would require additional nontrivial search and comparison, but this is indeed a promising direction and potentially an alternative or complementary approach to what is described in current work.
> > > 2. We agree that “informative” can be vague. In practice, we should consider features are potentially meaningful and worthy of further investigation, only when the observed shift corresponding to statistically significant impact due to model finetuning, e.g. performance improvement, efficiency gain. Thus, any “meaningful shifts” should be concluded with these specific contexts.

---

### Author Rebuttal · Authors · 2026-01-24

**Rebuttal:**

We thank all reviewers' constructive comments and attach an updated version of manuscript with modifications highlighted in cyan.

**Supporting Material:**

/attachment/7e4b68193a6bf6a97a767104050f2ba2287592c8.pdf

---

### Meta-Review · Area_Chair_ej9K · 2026-02-09

**Recommendation:** Accept (Poster)
**Confidence:** 4

**Metareview:**

Thank you to the authors and reviewers for their thoughtful discussion. Overall, the reviewers found this work to be addressing an interesting and important problem and producing some insightful findings. The major weaknesses highlighted by the reviewers were that only a single foundation model was used, questioning the generalizability of the findings and that formal, statistical analyses were not conducted. The authors were responsive to most other comments raised by the reviewers and the paper is improved as a result.

---

### Decision · Program_Chairs · 2026-02-13

Accept (Poster)